# Association between dietary folate intake and HPV infection: NHANES 2005–2016

Shuo Jin[1,2], Fangxuan Lin[1,2], Liuqing Yang[1,2,3], Qin Zhang[1,2,3]*

**1** Department of TCM Gynecology, Hangzhou TCM Hospital Affiliated to Zhejiang Chinese Medical University, Hangzhou, Zhejiang Province, China, **2** Zhejiang Chinese Medical University, Binjiang District, Hangzhou, China, **3** Research Institute of Women's Reproductive Health, Zhejiang Chinese Medical University, Binjiang District, Hangzhou, China

* zhaqin01@163.com

## Abstract

### Background

Recent studies have established a correlation between folate levels and the incidence of cervical cancer. Given that Human Papillomavirus (HPV) infection is a primary etiological factor in the development of cervical cancer, the nature of the relationship between dietary folate intake and HPV infection remains an area of ongoing investigation.

### Methods

To investigate the association between dietary folate intake and HPV infection, this study utilized data from the National Health and Nutrition Examination Survey (NHANES) spanning from 2005 to 2016. Multivariate logistic regression analysis was employed to examine the potential associations. Furthermore, the use of restricted cubic splines (RCS) facilitated the exploration of any non-linear correlations. Additionally, subgroup analyses were used to explore this correlation in different populations.

### Results

The study encompassed a total of 6747 women aged between 18 and 59 years. For every one mcg increase in folate intake, the incidence of HPV infection is reduced by 1% (OR = 0.99, p<0.05). Besides, folate intake was categorized into quartiles as follows: Q1 (<211 mcg/day), Q2 (211–311 mcg/day), Q3 (311–448 mcg/day), and Q4 (>448 mcg/day). The adjusted odds ratios (OR) for the different folate levels were as follows: Q2: 0.94 (95% CI: 0.76–1.16), Q3: 0.84 (95% CI: 0.67–1.04), and Q4: 0.63 (95% CI: 0.49–0.81). The RCS analysis confirmed a nonlinear relationship between dietary folate intake and HPV infection risk. Notably, a significant inverse association was observed when dietary folate intake exceeded 193.847 mcg/day.

**Data Availability Statement:** The datasets produced and examined in the ongoing research can be accessed on the NHANES website: https://www.cdc.gov/nchs/nhanes/index.htm.

**Funding:** This research was financially supported by the Construction Program for National Famous Traditional Chinese Medicine Experts Inheritance Studio in 2022 (No.75 [2022] of Chinese Medicine People's Education Letter) and Natural Science Foundation of Zhejiang Province, (No. LQ23H270017). The funders had no role in study design, data collection and analysis, decision to publish, or preparation of the manuscript. And no additional external funding was received for this study.

**Competing interests:** All authors declared that there have no conflict of interest.

## Conclusions

In conclusion, the findings of this study indicate a negative association between dietary folate intake and the risk of HPV infection. This association demonstrates a nonlinear pattern, particularly evident at higher levels of folate consumption.

## Introduction

Human papillomavirus (HPV), classified under the papillomaviridae family, is a diminutive double-stranded DNA virus proficient in adapting to its host's environment, thereby evading immune detection [1]. HPV ranks as the second most common viral agent implicated incancer etiology [2]. Depending on their potential for causing cancer or precursor lesions, HPV is further divided as low or high risk. High-risk HPV(HR-HPV) infection is especially associated with gynecological cancers [3]. Currently, HPV vaccination represents the primary preventive strategy against HPV infection. However, this measure faces financial constraints in less affluent nations, and even in wealthier countries, HPV vaccine coverage remains suboptimal. HPV vaccination has been introduced in 78.6% of high-income and upper-middle-income countries, compared to 37.5% of low-income and lower-middle-income countries in 2022 [4], underscoring the need for alternative, cost-effective preventive methods.

Several studies have identified a relationship between dietary habits, particularly the consumption of fruits and vegetables, and the risk of cervical cancer and HPV infection [5]. Notably, increased dietary intake of certain antioxidants, including vitamins A, B2, E, and folate, has been associated with a reduced risk of HPV infection [6]. Folate, also known as vitamin B9, is an essential nutrient not synthesized endogenously in humans. Dietary sources such as meat, fish, dairy products, and cereals are therefore vital for its acquisition [7]. Emerging evidence suggests that bodily folate levels inversely correlate with the carcinogenesis process in HPV-associated cancers, possibly due to folate's role in maintaining a highly methylated state of the HPV genome [8, 9].

Existing studies have confirmed that HR-HPV infection can be carcinogenic, and folate is a beneficial preventative measure against HR-HPV and cervical cancer, so it is essential to conduct a comprehensive study to fully understand the correlation between HPV infection and dietary folate intake, with a view to making progress in the prevention of tumorigenesis and development. In our study, we found that higher dietary folate intake reduced the prevalence of HPV infection, indicating a dose-effect relationship, and this relationship remained significant in the majority of population subgroups. This result suggested that dietary folate supplementation is an effective method of preventing HPV infection.

## Methods

### Data sources and study population

Our study included NHANES participants from six 2-year cycles, from 2005–2016. As is well known, NHANES is a national survey performed by the National Center for Health Statistics (NCHS), which acquires a sample of the ambulatory U.S. population by random selection (each participant represents approximately 65,000 U.S. people), which combines interviews and physical examinations, both completed with professional assistance, to evaluate the health condition of the U.S. civilian. Its findings will be used to determine disease prevalence and risk factors. It helps to create effective public health regulations, guides and designs health

programs and services, and enhances the health literacy of citizens. The NHANES study proto-col has been granted approval by the NCHS Ethics Review Board (https://www.cdc.gov/nchs/nhanes/irba98.htm). Additionally, it is crucial that every participant in this study complete a written informed consent at the time of enrollment. The original protocol is linked online (accessed September 24, 2023) [10–12].

In the initial cohort of NHANES 2005–2016, a total of 30,784 women participated in our research. After a rigorous data cleansing process, we excluded individuals with missing data on covariates, dietary folate intake, HPV infection status. Of note, our study also excluded pregnant women as subjects, so the final analyzed dataset included 6,764 U.S. women. For a more complete understanding of the complexity of our study, please visit the official NHANES website link (access date: September 20, 2023) for more information. Fig 1 depicts the process of selecting participants.

## Assessment of dietary folate intake

NHANES participants were requested to record their dietary consumption in the 24 hours before the interview. As part of the study's data collection process, each NHANES participant received two separate 24-hour dietary recall interviews. The initial interview took place in Mobile Examination Center (MEC), while the subsequent interview was carried out three to ten days later through telephone interview. NHANES' comprehensive Dietary Investigator's Procedures Manual provides an extensive and detailed description of the examination proce-dures and techniques used to collect dietary information.

In our study, we calculated 24-hour dietary folate intake based on data obtained during the initial day of the dietary interview.

## Detection of HPV infections

HPV testing was performed on women between 18 and 59 years old. The vaginal swab is extracted to obtain HPV DNA. A comprehensive analysis of 37 distinct HPV genotypes was conducted using a complex linear array HPV genotyping assay from Roche Diagnostics. The study primarily aimed to ascertain the comprehensive HPV infection status of every partici-pant. If the HPV test results showed the absence of all 37 HPV types, we categorized this par-ticipant's HPV infection status as negative. Otherwise, the participant's HPV infection status was positive. In addition, to further complete the categorization, a positive test result for any HR-HPV types (HPV 16, 18, 31, 33, 35, 39, 45, 51, 52, 56, 58, and 59) was interpreted as high-risk HPV infection. Conversely, any other of the remaining HPV genotypes was considered a low-risk HPV infection.

## Covariates

Our study comprehensively explored the various factors that influenced our research. These factors included age, body mass index (BMI), poverty income ratio (PIR), dietary vitamin B6, vitamin B12, and folate intake, serum and red blood cell (RBC) folate levels, education degree, marital status, sexual behavior, and the participants' drinking and smoking habits.

Participants were categorized into different racial backgrounds, including White, Black and other races. Education degree was categorized as below high school and above high school. PIR was calculated according to established guidelines, taking into account household size, year, and state to produce an indicator of economic disparity. We describe the state of mar-riage in the following three ways: (1)never married, (2)married or living with a partner, (3) widowed, divorced or separated. Information on each nutrient was obtained from dietary data. Body mass index categories (underweight, normal, overweight and obese) [13] were

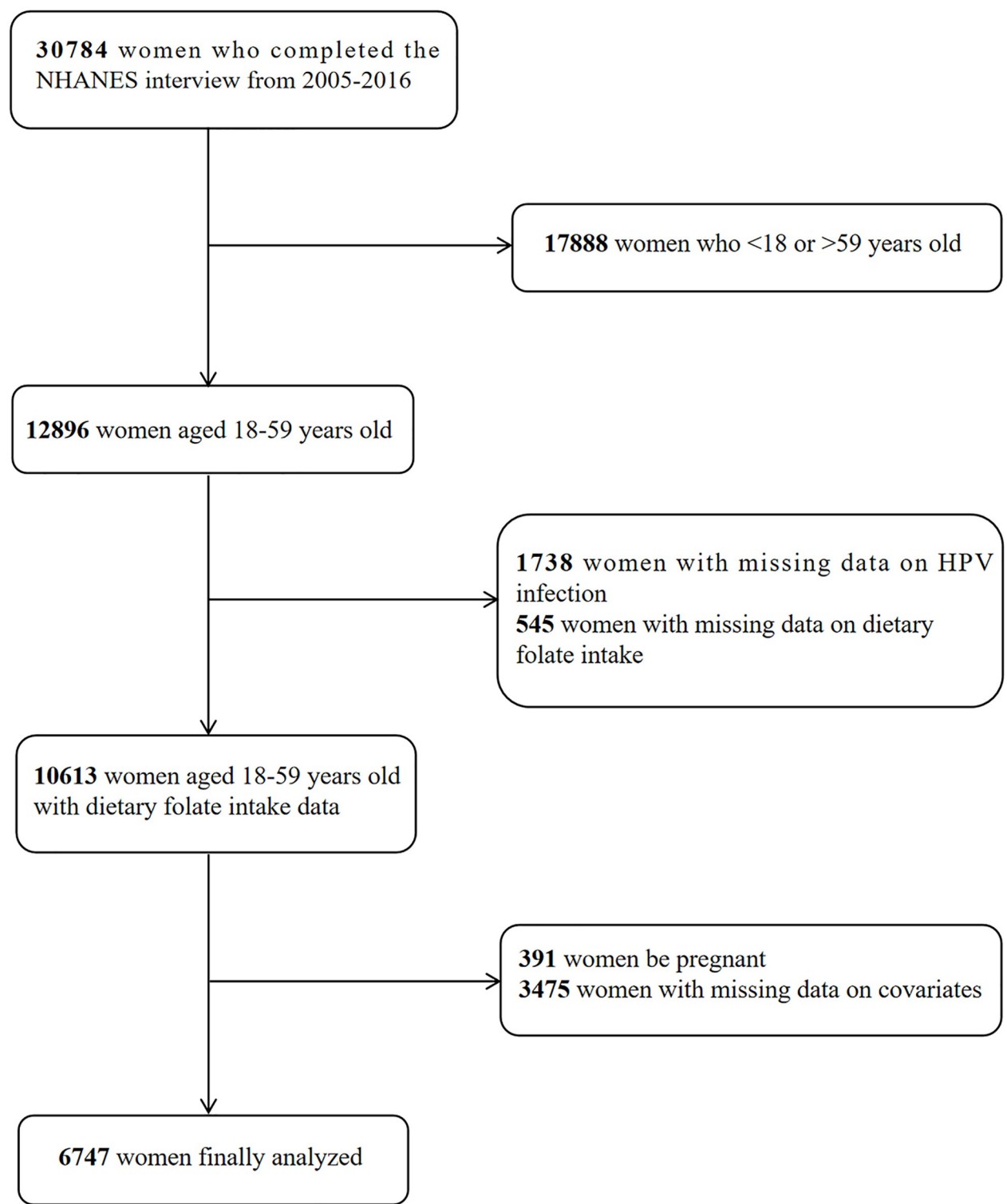

**Fig 1. Flowchart for choosing participants in HNANES from 2005–2016.**

derived from physical examination data. In addition, we collected information about participants' lifestyle habits through self-report questionnaires about their drinking and smoking and sexual behaviors. For drinking habits, we use the following definitions: (1) current heavy alcohol use: Includes people who drink three or more drinks per day and drink heavily (defined as more than four times drinks on the same occasion) five or more days per month, (2) current moderate alcohol use: Includes people who drink two or more drinks per day or who binge drink on two or more days per month,(3) current mild alcohol use: Includes people who drink one or more drinks per day, (4) former alcohol use: people who did not drink alcohol in the past year, but had twelve or more drinks in a year or over a lifetime,(5) never drink: Includes people who had fewer than twelve drinks in their lifetime [14]. For smoking, we categorized participants into three different categories:(1) now: People who smoke now and have smoked over 100 cigarettes.(2) former: People who have smoked over 100 cigarettes in their lifetime and have stopped smoking. (3) never: Total number of cigarettes smoked less than 100 [15].

## Statistical analysis

We applied means ± standard deviations (SD) to characterize the continuous variables. Frequencies and percentages were used to represent categorical variables. To complete the analyses, the R package 4.2.2 was applied. For continuous variables, we utilized the Kruskal-Wallis test to examine the differences in baseline characteristics in Table 1, and t-test or the Mann-Whitney U test was used to examine the differences in Table 2. For categorical variable, the chi-square test was applied.

The objective of our research was to explore the correlation between the consumption of dietary folate and HPV infection. To solve this problem, we utilized a multivariate logistic regression model. Model 1 was the original unadjusted model and Model 2 was adjusted for age and race. Model 3 was adjusted from Model 2 for additional covariates such as body mass index(BMI), poverty income ratio, education level, marital status, sexual behavior, vitamin B6 intake, vitamin B12 intake, dietary folate intake, serum folate, RBC folate, and alcohol and smoking status. $p$-value below 0.05 (bilaterally) defined as our standard for determining statistical significance. We employed restricted cubic spline curve (RCS) regression to discover the nonlinearity manner between folate intake and HPV infection.

## Results

A total of 30784 women finished the NHANES survey from 2005–2016. 12896 of the participants aged 18–59 were involved in our study. We excluded 1738 individuals with missing HPV testing results, 545 participants with missing data on dietary folate consumption, 3458 participants with missing covariates, and 391 pregnant. Ultimately, this study included 6747 participants in total. The average age was 38.17 years. The median level of dietary folate intake was 352.47 mcg/day. It was found that 2479 participants were infected with HPV.

We divided all participants into four groups according to the quartile of dietary folate intake (Q1, Q2, Q3, and Q4). Table 1 outlines the characteristics of the various folate consumption groups. According to the analysis, we found significant differences in different dietary folate intake groups. Demographic characteristics, such as race, marital status, and education degree ($p < 0.001$), have significant differences between different folate intake groups. Age and BMI differed significantly between groups (p < 0.05). In addition, HPV infection status was also significantly different in populations with different levels of dietary folate intake ($p < 0.001$), as do age at first sexual intercourse. The poverty ratio and smoking status also show a significant difference in folate intake groups. Serum folate and RBC folate differ significantly among

**Table 1. Stratification analysis for the association between dietary folate intake and HPV infection.**

| Characteristics | Dietary Folate Intake, mcg/d | | | | P-value |
|---|---|---|---|---|---|
| | Q1 (≤211 mcg) | Q2 (211–311 mcg) | Q3 (331–448 mcg) | Q4 (>448 mcg) | |
| Age (years) | 38.13(0.37) | 39.41(0.36) | 39.15(0.44) | 39.36(0.47) | 0.02 |
| BMI (kg/m2) | 29.07(0.26) | 28.71(0.26) | 28.33(0.24) | 28.20(0.27) | 0.03 |
| Poverty Income Ratio | 2.72(0.07) | 3.06(0.06) | 3.06(0.07) | 3.25(0.07) | < 0.0001 |
| Serum Folate (ng/ml) | 15.26(0.41) | 16.93(0.32) | 18.74(0.38) | 21.54(0.53) | < 0.0001 |
| RBC Folate (ng/ml) | 430.82(7.77) | 455.37(7.37) | 490.50(8.36) | 526.94(9.68) | < 0.0001 |
| VitaminB6 (mcg) | 1.13(0.04) | 1.55(0.03) | 1.85(0.03) | 2.55(0.04) | < 0.0001 |
| VitaminB12 (mg) | 2.77(0.08) | 3.43(0.06) | 4.34(0.09) | 6.46(0.28) | < 0.0001 |
| Race | | | | | < 0.0001 |
| White | 687(65.95%) | 716(66.25%) | 700(65.24%) | 733(68.42%) | |
| Black | 437(14.65%) | 385(12.94%) | 323(11.07%) | 267 (8.21%) | |
| Others | 576(19.4%) | 582(17.81%) | 677(23.69%) | 681(23.36%) | |
| Marital | | | | | < 0.0001 |
| Never married | 446(23.79%) | 358(17.07%) | 348(18.83%) | 352(16.76%) | |
| Married/Living with partner | 964(59.89%) | 1084(67.98%) | 1124(69.74%) | 1142(73.31%) | |
| Widowed/Divorced/Separated | 290(16.32%) | 241(14.96%) | 228(11.43%) | 187(9.93%) | |
| Education | | | | | < 0.0001 |
| Middle and high school | 818(43.16%) | 669(33.87%) | 617(31.17%) | 575(26.56%) | |
| College and higher | 882(56.84%) | 1014(66.13%) | 1083(68.83%) | 1106(73.44%) | |
| Age at first sexual intercourse (years) | | | | | 0.002 |
| No Sex | 96(3.78%) | 88(3.70%) | 90(4.21%) | 64(2.91%) | |
| <12 | 61(3.39%) | 54(2.73%) | 44(2.14%) | 39(1.86%) | |
| 13–18 | 948(59.00%) | 925(56.99%) | 888(53.27%) | 854(52.08%) | |
| 18–30 | 584(33.37%) | 594(35.72%) | 663(39.62%) | 711(42.79%) | |
| 30–40 | 8(0.25%) | 20(0.80%) | 12(0.72%) | 12(0.29%) | |
| >40 | 3(0.21%) | 2(0.06%) | 3(0.04%) | 1(0.07%) | |
| Number of sexual intercourse past year | | | | | 0.09 |
| No Sex | 96(3.78%) | 88(3.70%) | 90(4.21%) | 64(2.91%) | |
| Once | 78(3.14%) | 53(2.18%) | 61(3.11%) | 59(2.78%) | |
| 2–11 | 440(24.45%) | 376(22.23%) | 364(20.70%) | 360(20.69%) | |
| 12–51 | 509(31.36%) | 574(36.33%) | 599(36.97%) | 596(36.92%) | |
| 52–103 | 317(20.45%) | 341(19.78%) | 351(21.98%) | 359(22.58%) | |
| 104–364 | 237(15.25%) | 236(15.25%) | 218(12.38%) | 221(13.03%) | |
| >365 | 23(1.57%) | 15(0.52%) | 17(0.65%) | 22(1.10%) | |
| Unprotected sex | | | | | 0.3 |
| No Sex | 96(3.78%) | 88(3.70%) | 90(4.21%) | 64(2.91%) | |
| Never | 395(19.13%) | 380(19.70%) | 411(19.94%) | 398(20.39%) | |
| About half of the time | 444(24.63%) | 371(19.55%) | 409(21.12%) | 398(20.62%) | |
| Always | 765(52.45%) | 844(57.05%) | 790(54.73%) | 821(56.09%) | |
| Smoking status | | | | | < 0.0001 |
| Never | 1012(53.43%) | 1075(61.00%) | 1132(65.14%) | 1140(63.31%) | |
| Former | 217(16.44%) | 222(14.94%) | 234(14.97%) | 264(20.01%) | |
| Now | 471(30.13%) | 386(24.06%) | 334(19.89%) | 277(16.68%) | |
| Drinking status | | | | | 0.06 |
| Never | 267(11.64%) | 253(12.17%) | 252(11.47%) | 277(11.97%) | |
| Former | 214(11.53%) | 187(10.01%) | 201(11.52%) | 174(10.43%) | |
| Mild | 395(22.58%) | 432(27.02%) | 468(30.34%) | 440(29.12%) | |

*(Continued)*

**Table 1.** (Continued)

| Characteristics | Dietary Folate Intake, mcg/d | | | | P-value |
|---|---|---|---|---|---|
| | Q1 (≤211 mcg) | Q2 (211–311 mcg) | Q3 (331–448 mcg) | Q4 (>448 mcg) | |
| Moderate | 404(26.65%) | 402(25.95%) | 383(23.71%) | 407(26.38%) | |
| Heavy | 420(27.60%) | 409(24.84%) | 396(22.96%) | 383(22.09%) | |

Variables are presented as mean (SD) or n (%). Kruskal-Wallis test for continuous variables and Chi-square tests for categorical variables.

different folate intake groups. Meanwhile, vitamin B6 and vitamin B12 vary significantly based on folate intake levels.

Based on participants' HPV infection status, we display their characteristics in Table 2. Our study included 6747 individuals, and 2479 of them got HPV infection. The average age of HPV-infected women was 36.78 years, significantly younger than those without HPV infection (40.17, $p < 0.0001$). Additionally, participants without HPV infection had a higher PIR than those with HPV infection(3.22±0.05 vs. 2.66±0.06, $p < 0.001$). Serum folate, RBC folate, and dietary folate intake show significant differences between the two groups. Race, education, marital status, first sexual intercourse age, alcohol consumption, and smoking status also show a significant difference between women with or without HPV infection.

In our research, we constructed three models to confirm the connection between dietary folate consumption and HPV infection. Model 1 is the unadjusted model, and for every one mcg increase in folate intake, the incidence of HPV infection is reduced by 1%(OR = 0.99, $p<0.001$), after adjusting for all covariates in Model 3, this negative association remained stable.

In addition, we converted dietary folate intake to a categorical variable based on quartiles and found participants in the fourth quartile of folate consumption level were less likely to get HPV infection than those in first quartile in Model 3 (Q4 vs Q1, OR = 0.63,95% Cl:(0.49,0.81), $p<0.001$). We explored the association between folate intake and HPV genotype in an HPV-infected population but found no significant correlation. All models supported this conclusion. We show the result in Table 3.

After adjusting all covariates with a restricted cubic spline (RCS), we discovered a nonlinear correlation between folate intake and HPV infection (nonlinear $p = 0.03$). And we show the result in Fig 2.The inflection point was 193.85 mcg of dietary folate intake. At the range of 0–193.85mcg of folate intake, this association was insignificant (OR:1.00, $p = 0.55$). However, when folate intake exceeds 193.847mcg, we observed HPV infection risk reduced significantly. (OR:0.99, $p = 0.02$).

We analyzed subgroups by different factors, including age, BMI, race, marital status, smoking and drinking status and displayed the results in Table 4. We found interaction between dietary folate consumption and age ($p<0.001$). The OR in women aged 29–39 was 0.99 (0.99,1.00), 39–49 was 0.99 (0.99,1.00) and 49–59 was 0.99 (0.99,1.00). Besides, we also found that the OR between folate intake levels and HPV infection was more prominent in overweight (0.99 (0.99,1.00)) and obese (0.99 (0.99,1.00)) women. And only in Mexican women, the association is insignificant compared to other races, the OR in Black was 0.99 (0.99,1.00), in White was 0.99 (0.99,1.00), and other races was 0.99 (0.99,1.00). For women who have never smoked, the OR was 0.99 (0.99,1.00). For women who never drink the OR was 0.99 (0.99,1.00), for former drinkers the OR was 0.99 (0.99,1.00). For women who get married or living with partner, the OR was 0.99 (0.99,1.00).

**Table 2. Characteristics of participants according to HPV infection status(n = 6747).**

| Characteristics | HPV infection | | P-Value |
| --- | --- | --- | --- |
| | Yes | No | |
| Age (years) | 40.17(0.28) | 36.78(0.39) | < 0.0001 |
| BMI (kg/m2) | 28.54(0.17) | 28.63(0.19) | 0.65 |
| Poverty Income Ratio | 3.22(0.05) | 2.66(0.06) | < 0.0001 |
| Serum Folate | 1.79(0.03) | 1.78(0.03) | 0.86 |
| RBC Folate | 4.30(0.11) | 4.28(0.08) | 0.87 |
| Vitamin B6 (mcg) | 18.99(0.28) | 16.61(0.36) | < 0.0001 |
| Vitamin B12 (mg) | 489.94(7.27) | 451.42(5.54) | < 0.0001 |
| Dietary folate intake(mcg) | 371.82(5.44) | 336.08(5.87) | < 0.0001 |
| Race | | | < 0.0001 |
| White | 1865(68.82%) | 971(61.98%) | |
| Black | 706 (8.69%) | 706(17.45%) | |
| Others | 1714(22.5%) | 802(20.57%) | |
| Marital | | | < 0.0001 |
| Never married | 716(14.23%) | 788(28.47%) | |
| Married/Living with partner | 3130(76.28%) | 1184(51.36%) | |
| Widowed/Divorced/Separated | 439(9.49%) | 507(20.17%) | |
| Education | | | < 0.0001 |
| Middle and high school | 1601(30.82%) | 1078(38.84%) | |
| College and higher | 2684(69.18%) | 1401(61.16%) | |
| Age at first sexual intercourse (years) | | | < 0.0001 |
| No Sex | 276(4.83%) | 62(1.28%) | |
| <12 | 101(2.00%) | 97(3.52%) | |
| 13–18 | 2057(50.17%) | 1558(65.28%) | |
| 18–30 | 1805(42.27%) | 747(29.54%) | |
| 30–40 | 41(0.62%) | 11(0.29%) | |
| >40 | 5(0.10%) | 4(0.08%) | |
| Number of sexual intercourse past year | | | < 0.0001 |
| No Sex | 276(4.83%) | 62(1.28%) | |
| Once | 152(2.59%) | 99(3.21%) | |
| 2–11 | 918(21.06%) | 622(23.81%) | |
| 12–51 | 1464(36.07%) | 814(34.20%) | |
| 52–103 | 881(21.40%) | 487(20.85%) | |
| 104–364 | 551(13.26%) | 361(15.33%) | |
| >365 | 43(0.78%) | 34(1.31%) | |
| unprotectedsex | | | < 0.0001 |
| No Sex | 276(4.83%) | 62(1.28%) | |
| Never | 1038(20.34%) | 546(18.75%) | |
| About half of the time | 786(16.15%) | 836(31.83%) | |
| Always | 2185(58.67%) | 1035(48.15%) | |
| Smoking status | | | < 0.0001 |
| Never | 2934(64.13%) | 1425(54.23%) | |
| Former | 632(18.07%) | 305(13.89%) | |
| Now | 719(17.80%) | 749(31.89%) | |
| Drinking status | | | < 0.0001 |
| Never | 781(14.02%) | 268(7.50%) | |
| Former | 500(11.07%) | 276(10.45%) | |

(*Continued*)

**Table 2.** (Continued)

| Characteristics | HPV infection | | P-Value |
|---|---|---|---|
| | Yes | No | |
| Mild | 1193(29.66%) | 542(22.71%) | |
| Moderate | 984(25.45%) | 612(26.17%) | |
| Heavy | 827(19.80%) | 781(33.18%) | |

Variables are presented as mean (SD) for continuous or n (%) for categorical. For continuous variables, t-test or the Mann-Whitney U test was used. For categorical variables, Chi-square tests was used.

## Discussion

Nowadays, HPV infection is a prevalent concern, with estimates suggesting that approximately 80% of women encounter HPV at some point in their lives. HR-HPV is particularly implicated in the carcinogenesis process. According to the 2020 Cervical Cancer Epidemiologic Survey, there were approximately 604,127 new cervical cancer cases and 341,831 deaths globally, ranking it as the fourth most common cancer among women. These figures underscore the ongoing challenges in achieving the WHO's targets for cervical cancer prevention and treatment [16]. To achieve the global public health goal of eliminating cervical cancer, WHO has proposed the "90-70-90" goal. Aiming for 90% of girls to be completely vaccinated against HPV by the age of 15, 70% of women undergo high-performance testing by 35, and by 45, 90% of women diagnosed with cervical disease receive treatment.

Our study involved data from the 2005–2016 NHANES database, including dietary folate consumption data and HPV infection data for 6747 U.S. women. After adjusting for covariates, we observed a consistent negative association between dietary folate intake and HPV

**Table 3.** Association between dietary folate intake and HPV infection status.

| | Model 1[a] | | Model 2[b] | | Model 3[c] | |
|---|---|---|---|---|---|---|
| | OR (95% CI) | P-Value | OR (95% CI) | P-Value | OR (95% CI) | P-Value |
| HPV vs. No HPV | | | | | | |
| Dietary Folate Intake(mcg/day) | 0.99 (0.99–1.00) | <0.0001 | 0.99 (0.99–1.00) | <0.001 | 0.99 (0.99–1.00) | 0.004 |
| Q1 (< 211) | Ref | | Ref | | Ref | |
| Q2 (211–311) | 0.83 (0.69,1.00) | 0.04 | 0.86(0.71,1.04) | 0.12 | 0.94 (0.76,1.16) | 0.53 |
| Q3 (311–448) | 0.72 (0.59,0.88) | 0.001 | 0.75(0.61,0.92) | 0.01 | 0.84 (0.67,1.04) | 0.11 |
| Q4 (> 448) | 0.56 (0.46,0.69) | <0.0001 | 0.60(0.49,0.74) | <0.0001 | 0.63 (0.49,0.81) | <0.001 |
| HR-HPV vs. LR-HPV | | | | | | |
| Dietary Folate Intake(mcg/day) | 1.00 (0.99–1.00) | 0.97 | 0.99 (0.99–1.00) | 0.67 | 0.99 (0.99–1.00) | 0.45 |
| Q1 (< 211) | Ref | | Ref | | Ref | |
| Q2 (211–311) | 1.14(0.81,1.60) | 0.44 | 1.17(0.83,1.66) | 0.35 | 1.12(0.80,1.56) | 0.51 |
| Q3 (311–448) | 1.06(0.77,1.45) | 0.74 | 1.02(0.74,1.42) | 0.87 | 1.02(0.74,1.40) | 0.92 |
| Q4 (> 448) | 1.06(0.77,1.45) | 0.73 | 0.99(0.72,1.37) | 0.97 | 0.94(0.64,1.38) | 0.75 |

[a] Model 1 was the unadjusted model.

[b] Model 2 was adjusted for age and race.

[c] Model 3 was adjusted for age, race, body mass index(BMI), poverty income ratio, education level, marital status, sexual behavior, vitamin B6 intake, vitamin B12 intake, dietary folate intake, serum folate, RBC folate, and alcohol and smoking.

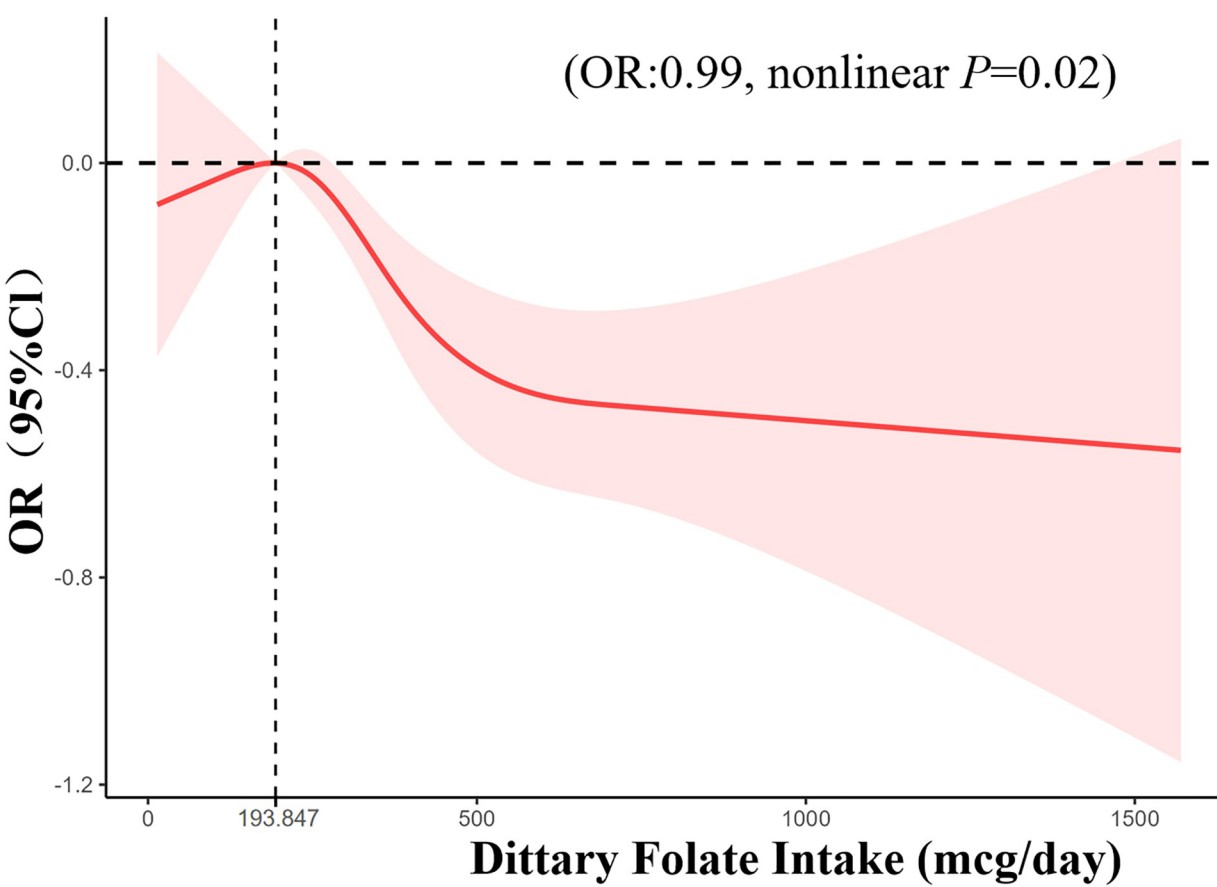

**Fig 2. Nonlinear relationship between dietary folate intake and HPV infection.**

infection. Additionally, no significant association was found between different HPV genotypes and dietary folate intake. The RCS method was employed to investigate the non-linear relationship, revealing a decrease in HPV infection risk with dietary folate intake exceeding 193.85 mcg/day.

So far, some researchers have noticed the connection between dietary folate consumption and HPV infection. One study of 485 multiethnic women reported the negative correlation between the consumption of folate and squamous intraepithelial lesions, this association is evident at both low-grade and high-grade manifestations [17].

Additionally, Asemi started a randomized, double-masked study and discovered women who consistently supplemented with folic acid had a significant decrease in CIN1. The women in the treatment group showed an improvement rate of 83.3%, which was significantly higher than the 52.0% observed in the control group [18]. As for the relationship between folate and HPV genotype, a clinical trial reported patients with various forms of HPV did not significantly differ in their folate levels [19], which shows the same conclusion with our study.

Since folate is a vitamin that the body cannot produce on its own. It is usually derived from food such as meat, fish, vegetables and so on [20]. Folate is essential for the metabolism of 1-carbons, which is connected to DNA methylation [21]. Deficiency in folate not only affects

**Table 4. Stratification Analysis for the association between dietary folate intake and HPV infection.**

| Characteristics | N | OR (95% CI) | P-Value | P for Interaction |
|---|---|---|---|---|
| Age | | | | <0.001 |
| <29 | 1821 | 1.00 (0.99–1.00) | 0.721 | |
| 29–39 | 1789 | 0.99 (0.99–1.00) | 0.036 | |
| 39–49 | 1868 | 0.99 (0.99–1.00) | 0.003 | |
| 49–59 | 1269 | 0.99 (0.99–1.00) | <0.001 | |
| BMI | | | | 0.225 |
| Underweight | 145 | 0.99 (0.99–1.00) | 0.097 | |
| Normal Weight | 1536 | 1.00 (0.99–1.00) | 0.19 | |
| Overweight | 3335 | 0.99 (0.99–1.00) | 0.048 | |
| Obese | 1731 | 0.99 (0.99–1.00) | <0.001 | |
| Race | | | | 0.611 |
| White | 2828 | 0.99 (0.99–1.00) | 0.004 | |
| Black | 1408 | 0.99 (0.99–1.00) | 0.005 | |
| Others | 1322 | 0.99 (0.99–1.00) | 0.097 | |
| Marital | | | | 0.034 |
| Never married | 1498 | 1.00 (0.99–1.00) | 0.597 | |
| Married/Living with partner | 4305 | 0.99 (0.99–1.00) | <0.001 | |
| Widowed/Divorced/Separated | 944 | 1.00 (0.99–1.00) | 0.818 | |
| Smoking status | | | | 0.592 |
| Never | 4349 | 0.99 (0.99–1.00) | 0.002 | |
| Former | 934 | 0.99 (0.99–1.00) | 0.065 | |
| Now | 1464 | 1.00 (0.99–1.00) | 0.292 | |
| Drinking status | | | | 0.319 |
| Never | 1044 | 0.99 (0.99–1.00) | 0.027 | |
| Former | 774 | 0.99 (0.99–1.00) | 0.004 | |
| Mild | 1731 | 0.99 (0.99–1.00) | 0.096 | |
| Moderate | 1593 | 0.99 (0.99–1.00) | 0.068 | |
| Heavy | 1605 | 1.00 (0.99–1.00) | 0.256 | |

DNA damage but also DNA repair [22]. The deficiency of folate is connected with the increasing risk of cancer [23], cardiovascular disease [24], and neural tube defects [25].

The persistent presence of HR-HPV infection significantly contributes to the progression of cervical cancer. One research found about 99% detection rate of HR-HPV in cervical cancer samples [26]. HPV integration in the genome causes the functional deficiency of tumor suppressor genes and genomic instability, which is one of the essential mechanisms of cervical cancer [27]. Additionally, HPV infections result in persistent inflammation and the release of cytokines, potentially influencing the expression of genes that may contribute to cancer [28].

Folate is involved in a variety of methylation reactions, thereby preventing the proliferation and persistence of virus [29]. Research has shown that low folate metabolic stress reprograms DNA methylation to promote epithelial-mesenchymal transition, and increase the risk of cervical cancer [30]. Folate deficiency can disrupt the cell cycle and affect the development of cervical cancer [31]. Thus, inadequate dietary intake of folate may be a causative factor for HPV infection. Supplementation of folate is an effective method to prevent cervical cancer. In addition, demethylation therapy may provide new strategies for cervical cancer treatment.

This study has elucidated a direct association between dietary folate consumption and socio-economic indicators, specifically family income and educational attainment. Notably, our analysis revealed significant racial disparities in folic acid intake, with African-American populations exhibiting lower levels compared to other racial groups. This discrepancy may be attributed to variations in lifestyle choices and socio-economic status. Furthermore, our findings indicate a correlation between marital status and dietary folate intake, where women who are married or cohabiting tend to have higher folate consumption. Intriguingly, an inverse relationship was observed between early sexual initiation and folate levels, suggesting potential behavioral or educational factors at play. The study also underscores the critical role of social determinants in the prevalence of HPV [32]. Lifestyle factors, including smoking and alcohol consumption, were identified as contributing to the increased susceptibility to HPV. The use of condoms emerged as a significant preventive measure against HPV infection, highlighting the importance of sexual health education in mitigating the spread of this virus.

We employed the restricted cubic spline to depict the non-linear correlation between dietary folate consumption and HPV infection. The curve revealed an inverse relationship between the consumption of dietary folate and HPV infection, which is consistent with previous research. Apart from these, we found some differences. The existence of this negative association requires some conditions, only when the dietary folate intake is >193.847mcg/day. The curve shows when dietary folate intake is <193.847mcg/day, there seems to be a positive correlation between folate levels and HPV infection status, although the conclusion was not significant, we still need further research.

Our research has the following strengths. First, our study focused on the currently unclear connection between dietary folate intake and HPV infection status, and applied subgroup analyses to explore the relationship between different subtypes of HPV. Additionally, we utilized the data obtained from a diverse national sample of 6747 American women gathered over six NHANES cycles, resulting in a comprehensive dataset. Furthermore, we employed stringent statistical techniques, such as multiple logistic regression and subgroup analysis, to modify covariates and investigate the correlations between folate consumption and HPV infection. Besides, we used a restricted cubic spline to explore the nonlinear relationship and found the presence of dose-response relationship.

We also have several drawbacks. First, as a cross-sectional study, it's unable for us to obtain the causality of dietary folate intake and HPV infection from our research. Second, we obtained the dietary folate intake data from one 24-hour dietary recall, which may have a recall bias. Thirdly, our study included the main covariates related to HPV infection but not all related covariates. Finally, there are some unexplainable relationships between covariables and processing variables in logical regression. Further cohort studies or randomized controlled trials are desperately needed to confirm this finding so that more precise and potent HPV infection prevention and treatment options may be offered.

## Author Contributions

**Conceptualization:** Shuo Jin.

**Data curation:** Shuo Jin, Fangxuan Lin.

**Supervision:** Qin Zhang.

**Writing – original draft:** Shuo Jin, Fangxuan Lin, Liuqing Yang.

**Writing – review & editing:** Fangxuan Lin, Liuqing Yang.

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
