## [Decision Letter · Decision Letter 0]

9 May 2024

PONE-D-24-04647Negative association between dietary folate intake and HPV infection: NHANES 2005–2016PLOS ONE

Dear Dr. Zhang,

Thank you for submitting your manuscript to PLOS ONE. After careful consideration, we feel that it has merit but does not fully meet PLOS ONE’s publication criteria as it currently stands. Therefore, we invite you to submit a revised version of the manuscript that addresses the points raised during the review process.

We look forward to receiving your revised manuscript.

Kind regards,

Kehinde S. Okunade

Academic Editor

PLOS ONE

Journal Requirements:

This research was financially supported by the Construction Program for National Famous Traditional Chinese Medicine Experts Inheritance Studio in 2022 (No.75 [2022] of Chinese Medicine People's Education Letter) and Zhejiang Provincial Natural Science Foundation of China (No.LQ23H270017).

Reviewers' comments:

Reviewer's Responses to Questions

**Comments to the Author**

1. Is the manuscript technically sound, and do the data support the conclusions?

Reviewer #1: Yes

Reviewer #2: Yes

2. Has the statistical analysis been performed appropriately and rigorously? 

Reviewer #1: Yes

Reviewer #2: No

3. Have the authors made all data underlying the findings in their manuscript fully available?

Reviewer #1: Yes

Reviewer #2: No

4. Is the manuscript presented in an intelligible fashion and written in standard English?

Reviewer #1: Yes

Reviewer #2: Yes

5. Review Comments to the Author

**Reviewer #1:** The topic of the study needs to be rephrased to show the work that has been done. The current topic is a rather the outcome of the research. The study is a good one with some incorrect information and grammatical mistakes that has been highlighted in the review attached. The author needs to recheck some information in the study to align with current evidence on HPV and cervical cancer. The objectives of this study were also not clearly stated.

**Reviewer #2: **Title and Abstract:

The title is concise and informative.

The abstract adequately summarizes the study's background, methods, results, and conclusions.

However, it lacks specific numerical results, which could be included to provide a clearer picture of the findings.

Introduction:

The introduction provides a good overview of HPV, its association with cervical cancer, and the importance of dietary factors, especially folate, in preventing HPV infection.

However, it could benefit from more recent citations to strengthen the relevance of the study.

The introduction lacks a clear statement of the study's objectives, which would help readers understand the research focus better. This section instead concludes by telling us the study finding instead of the study focus and aim.

Methods:

The methods section is detailed and well-organized, providing clear explanations of data sources, study population, variables, and statistical analyses.

However, there are minor grammatical errors and inconsistencies in the writing that could be addressed for clarity.

Results:

The results are presented clearly, with appropriate tables summarizing participant characteristics and statistical analyses.

However, some tables lack descriptive terms or are formatted inconsistently, which could make them challenging to interpret. (e.g. it is not clear if the Age in most tables are mean or median, the statistical tests applied are not stated especially in table 2 )

In some table, units of variables were omitted.

Discussion:

The discussion provides a thorough analysis of the study's results in the context of existing literature.

However, it could be strengthened by discussing potential limitations of the study, such as selection bias, measurement error, or confounding factors.

Additionally, suggestions for future research directions could be included to guide further investigation in this area.

Overall:

The manuscript presents valuable research on the association between dietary folate intake and HPV infection, with clear methodology and results.

However, there are areas for improvement in terms of clarity, organization, and interpretation of results, as well as addressing minor grammatical errors throughout the text.

6. PLOS authors have the option to publish the peer review history of their article (what does this mean?). If published, this will include your full peer review and any attached files.

Reviewer #1: No

Reviewer #2: No

---

## [Author Response · Author response to Decision Letter 0]

10 Jun 2024

Point-to-point responses to the comments:

Suggestions from editor:

This research was financially supported by the Construction Program for National Famous Traditional Chinese Medicine Experts Inheritance Studio in 2022 (No.75 [2022] of Chinese Medicine People's Education Letter) and Zhejiang Provincial Natural Science Foundation of China (No.LQ23H270017).

Response：

Thank you for your suggestions. We have carefully checked the format of the manuscript. Besides, We have revised the Funding Statement. (Line 392-394, Page 19).

Suggestions from reviewers:

Reviewer #1: 

The topic of the study needs to be rephrased to show the work that has been done. The current topic is a rather the outcome of the research. The study is a good one with some incorrect information and grammatical mistakes that has been highlighted in the review attached. The author needs to recheck some information in the study to align with current evidence on HPV and cervical cancer. The objectives of this study were also not clearly stated.

Response：

We apologize for the poor language of our manuscript. We tried our best to improve the manuscript and made some changes to the manuscript. These changes will not influence the content and framework of the paper. We appreciate for Reviewers’ warm work earnestly and hope that the correction will meet with approval.

Regarding our misleading title, we revised it as ’Association between dietary folate intake and HPV infection: NHANES 2005–2016’, hoping this correction avoids any misleading. (Line 1, Page 1).

We research the background of the research and revise the HPV vaccination rate based on recent statistics.(Line 53-55, Page 3).

In addition, we have revised the racial categorization into white, black and other races(Line 117, Page 6) and re-analyze the relevant data in Table 1-4.

We modified the categorization of BMI as underweight, normal, and overweight(Line 122, Page 6) to correspond to the later phase.

The expression “Condomless sex” is unclear, and we modify it to “Unprotected sex” according to the reviewers' comments in Table 1 and Table 2.

We've revised the incorrect expression “categorization” as “categorical variable”. (Line 140, Page 7)

We also described the current status of HPV prevention and supplemented it with the ‘90,70,90’ strategy of WHO. (Line 219-222, Page 14)

Thanks again for your constructive feedback.

Reviewer #2: 

Title and Abstract:

The title is concise and informative.

The abstract adequately summarizes the study's background, methods, results, and conclusions.

However, it lacks specific numerical results, which could be included to provide a clearer picture of the findings.

Introduction:

The introduction provides a good overview of HPV, its association with cervical cancer, and the importance of dietary factors, especially folate, in preventing HPV infection.

However, it could benefit from more recent citations to strengthen the relevance of the study.

The introduction lacks a clear statement of the study's objectives, which would help readers understand the research focus better. This section instead concludes by telling us the study finding instead of the study focus and aim.

Methods:

The methods section is detailed and well-organized, providing clear explanations of data sources, study population, variables, and statistical analyses.

However, there are minor grammatical errors and inconsistencies in the writing that could be addressed for clarity.

Results:

The results are presented clearly, with appropriate tables summarizing participant characteristics and statistical analyses.

However, some tables lack descriptive terms or are formatted inconsistently, which could make them challenging to interpret. (e.g. it is not clear if the Age in most tables are mean or median, the statistical tests applied are not stated especially in table 2 )

In some table, units of variables were omitted.

Discussion:

The discussion provides a thorough analysis of the study's results in the context of existing literature.

However, it could be strengthened by discussing potential limitations of the study, such as selection bias, measurement error, or confounding factors.

Additionally, suggestions for future research directions could be included to guide further investigation in this area.

Overall:

The manuscript presents valuable research on the association between dietary folate intake and HPV infection, with clear methodology and results.

However, there are areas for improvement in terms of clarity, organization, and interpretation of results, as well as addressing minor grammatical errors throughout the text.

Response：

1.For “Title and Abstract”, we examined the relationship between each mcg folate intake and the risk of HPV infection, such as for every one mcg increase in folate intake, the incidence of HPV infection is reduced by 1% (Line 32-33, Page 2). We hope that this part of the data will make our study more convincing.

2.For “Introduction”, we cited the more recent studies to describe the current global situation in the prevention and treatment of HPV (Line 54-55, Page 3). In addition, we have added the findings of our study in this section to help readers understand the focus and aim of this study. (Line 67-70, Page 4)

3.For “Methods”, we have tried our best to improve the language and grammar in the revised manuscript. (Line 113, Page 6; Line 137, Page 7)

4.For “Results”, thank you for your kind suggestions and we have modified the presentation of the data in Table1 and Table2. Besides, variables are presented as mean (SD) for continuous or n (%) for categorical in our research. In addition, the Kruskal-Wallis test was used to examine continuous variables in Table 1, while the t-test or the Mann-Whitney U test was used to examine the differences in Table 2. Besides, the chi-square test was applied for categorical variables. (Line 139-140, Page 7)

5.For “Discussion”, we added future research directions in this part. To further investigate the relationship between folate intake and HPV infection, cohort studies or randomized controlled trials are desperately needed. (Line 289-291, Page 17)

Thanks again for your insightful comments and kind suggestions.

We tried our best to improve the manuscript and made some changes in the revised paper which will not influence the content and framework of the paper. We appreciate for Editors/Reviewers’ warm work earnestly and hope the correction will meet with approval. Once again, thank you very much for your comments and suggestions.

---

## [Editor Report · Decision Letter 1]

21 Jun 2024

Association between dietary folate intake and HPV infection: NHANES 2005–2016

PONE-D-24-04647R1

Dear Dr. Zhang,

We’re pleased to inform you that your manuscript has been judged scientifically suitable for publication and will be formally accepted for publication once it meets all outstanding technical requirements.

Kind regards,

Kehinde S. Okunade

Academic Editor

PLOS ONE

Additional Editor Comments (optional):

Accept manuscript